# Well-Being and Stress of Children in Teaching by Digital Means during the COVID-19 Pandemic: A Case in Santarém, Brazilian Amazonia

**DOI:** 10.3390/ijerph19127148

**Published:** 2022-06-10

**Authors:** Kaio Vinícius Paiva Albarado, Iani Dias Lauer-Leite, Dennison Célio de Oliveira Carvalho, Thiago Almeida Vieira

**Affiliations:** 1Faculty of Medicine, Federal University of Pará, Altamira 68372-191, Brazil; 2Postgraduate Program in Society, Environment and Quality of Life (PPGSAQ), Federal University of Western Pará, Santarém 68040-255, Brazil; iani.leite@ufopa.edu.br; 3Postgraduate Program Education (PPGE), Federal University of Western Pará, Santarém 68040-328, Brazil; 4Centre for Interdisciplinary Formation, Federal University of Western Pará, Santarém 68040-255, Brazil; dennison.carvalho@ufopa.edu.br

**Keywords:** subjective well-being, quality of life, coronavirus, SARS-CoV-2, Brazil

## Abstract

The COVID-19 pandemic has caused and is still causing many infections. An important change brought about by prevention actions was the closing of schools, and the adoption of teaching by digital means in private institutions. In this article, we aim to analyze the subjective well-being and stress of children aged 8 to 12 years in digital education during the COVID-19 pandemic in a city in the Brazilian Amazon. For data collection, we used the Children’s Global Life Satisfaction Scale, the Multidimensional Scale of Life Satisfaction for Children (MSLSC), and the Infantile Stress Scale, all via Google Forms. To analyze the data, we used descriptive statistics, correlation between variables and logistic regression. The Family domain obtained higher scores (4.54 ± 0.45). The Infantile Stress Scale showed that the children were in a normal phase, with some of them in an alert and even resistance phase. There was a strong positive correlation between the Family domain (r = 0.70; *p*-value < 0.05) and the Self domain (r = 0.70; *p*-value < 0.001). The higher the value of the predictor variable (Family domain), the lower the chances of a child belonging to the Resistance category. Therefore, the Family domain is an important component of children’s well-being, acting as a protective factor against child stress.

## 1. Introduction

Throughout history, several large-scale public health problems have arisen. In the year 2019, in Wuhan, Hubei Province, China, a serious disease called COVID-19 (corona virus disease 2019) emerged [1]. This disease is responsible for cases of Severe Acute Respiratory Syndrome caused by coronavirus 2 (SARS-CoV-2), which put the world in the current pandemic situation. Consequently, there were changes in the routine of the population at a global level.

The World Health Organization (WHO) declared COVID-19 as a pandemic on 11 March 2020. Thus, due to the severity of COVID-19 infections, in 2020 the adoption of some non-pharmacological measures of individual scope began, such as frequent hand washing with soap and water, use of 70% alcohol, use of masks and social distancing, the latter in order to avoid places with large agglomerations, reducing the risk of exposure [2]. This health crisis generated negative problems in the mental health of the world population [3], such as anxiety problems in health professionals (36%), university students (34.7%), general population (34%), teachers (27.2%), parents (23.3%), pregnant women (19.5%) and police (8.79%) [4]. In addition, children were also affected, especially by the unprecedented change in routines and behaviors [5].

As one of the recommendations implemented to mitigate the proliferation and infection of the virus was social distancing, children spent more time at home. However, implementing social distancing with children has become one of the biggest obstacles in several countries in the face of different existing contexts [6].

In Brazil, each state took measures that were relevant to its contexts and times. In the Brazilian Amazonia, in the state of Pará, decrees prevented schools from operating in person, to prevent the spread of the disease. The adaptation of schools to this new reality varied: state and municipal public schools halted their activities. In private schools, strategies were diversified: at the beginning of the pandemic, some anticipated school holidays, followed by the implementation of remote activities, mediated by technological resources. Other schools in the private network implemented remote teaching immediately, through digital means. Still others chose to send assignments for students to carry out at home and later hand in at the institution. Common to the situation was the necessity of students remaining in their residence and the need for families to adapt to a series of changes in daily life.

When it comes to children, one of the main changes was not going to school and staying at home longer. A longer stay in a home environment, the absence of socializing with friends and the impossibility of going to other places (such as churches, squares, clubs, shopping malls, etc.) can generate stress, triggering symptoms of anxiety in children, impairing their development. In times of a pandemic, the concern and fear of infecting oneself and others become a source of stress for everyone [7,8].

In a survey of clinical manifestations and severity of children and adolescents treated at a hospital in the state of São Paulo, it was observed that in most cases, children had mild symptoms restricted to the upper airway [9]. Therefore, schools in the private network, through government decrees, were forced to hold classes remotely.

In this context, children in remote learning are or were in constant interaction via the internet with friends, schoolmates and family members. This community may have more contact with digital devices and put aside the practices of physical activities, bringing negative psychological effects due to the lack of face-to-face interaction and the prolonged period of sedentary lifestyle [10].

This scenario can bring damage to children’s subjective well-being (SWB), causing possible stress conditions during the COVID-19 pandemic due to the distance from the school environment and longer residential coexistence. SWB is characterized as the subject’s own assessment of life in an attempt to identify in which aspects he/she is “happy” with his/her own life and therefore, for many years, it was synonymous with happiness. Emotions such as sadness or anger, and episodes of anxiety and depression may be revealed according to individual experience [11,12].

Subjective well-being is a construct that involves two components: (a) the emotional component related to positive and negative affections and (b) the cognitive component related to life satisfaction [13,14]. In the first, the study of the emotional aspects of SWB aims to verify the balance between positive and negative life experiences. In the second, the subject makes his own judgment about how he feels about life [15]. With this, the emotional aspects of the individual are approached subjectively with aspects such as happiness, peace, fulfillment and satisfaction with life according to the analysis carried out by the subject’s own eyes [16].

The study on SWB presents two philosophical currents called Hedonic Welfare and Eudaemonic Welfare [17]. The hedonic aspect approaches the idea that well-being is linked to feelings that provide happiness or pleasure; the eudaemonic verifies the adequate functionality of the individual potential related to the ability to reason and common sense [15].

Therefore, the understanding of hedonism refers to the affective accumulation, with various emotions experienced by the subject throughout life, always concerned with knowing the “how” and “why” of people going through situations that can provide them with pleasure [18]. For these authors, the eudaemonic concept deals with the moment that happiness is generated, through the search for personal accomplishments.

Studies on SWB aim to understand the degree of personal satisfaction, usually through one- or multi-dimensional self-administered instruments. However, SWB research involving children can still be considered incipient compared with that of adults. In Brazil, there has been a growing development of scales that assess the components of children’s SWB [19].

The term “stress” was first used in 1926 by Dr. Hans Selye, describing it as an intense state of internal conflict in the subject exposed to some event outside its context [15]. In 1936, Dr. Hans Selye then defined stress as a state of defense of the organism in the face of unusual events experienced by human beings, also called General Adaptation Syndrome (GAS). This syndrome has three distinct phases: (a) alarm reaction; (b) resistance; and (c) exhaustion. In the first, the first contact with the stress-causing agent takes place. If contact with this event persists, the organism enters an adaptive process referring to the second stage. If the stress episode continues and there is no emotional control in the face of this situation, the individual will be in the third phase, with systemic physiological imbalance occurring, which can lead to death [20].

In the biological field, the third stage of GAS occurs with the secretion of the hormone that secretes corticotropin, which in turn will stimulate the pituitary gland to release the adrenocorticotropic agent, which acts on the adrenal gland, releasing cortisol [21].

Cortisol is a hormone belonging to the glucocorticosteroid class. Its synthesis occurs through steroid cholesterol. They are generated by the adrenal cortex and produce an average of 10 to 20 mg per day, and are considered one of the main corticosteroids in the body [22]. This hormone is involved in the stress response, increased heart pressure and glucose, as well as reducing the activity of the immune system in situations of inflammation; the blood concentration of cortisol has a half-life of 80–100 min [22]. With the increase in this hormone, changes in sleep pattern begin to occur more frequently, triggering other pathologies [23]. These changes occur within the biological field when faced with stressful situations.

The physiological response to a stressor stimulus of any nature, including those related to the psychological, is reflected in the physical organism. High levels of stress can affect the human body, triggering numerous diseases, including depression, cardiovascular diseases, gastric ulcers, appearance of herpes, and stroke, among others [24].

Every individual in the process of development can experience uncomfortable situations. The psychological factor, when directly affected, will present some disorder in the future. Thus, despite the various situations that an individual must face, the subject will suffer serious health problems, putting their biological integrity at risk [25].

To analyze possible changes in the SWB and stress of children during the COVID-19 pandemic, a study was conducted in India in which the effects of quarantine on the public were observed, directly affecting the psychological aspects of the participants [26]. In Brazil, studies that measured children’s SWB [27] found that the family has a direct influence on the increase in SWB levels. With this, studies that verify childhood SWB related to stress are necessary, mainly in the Amazonian context, to give answers regarding the pattern of occurrence of stress in children. Therefore, we verified the need to observe whether stress levels can influence SWB scores in children from a city in the Brazilian Amazon.

Due to the intense use of technologies for online studies, together with the prohibition of meeting with relatives and friends, the problem question arises: What is the state of subjective well-being and stress of children in education through digital means during the COVID-19 pandemic in a city in the Brazilian Amazon?

Therefore, we aimed to analyze the subjective well-being and stress of children aged 8 to 12 years in digital education during the COVID-19 pandemic in Santarém, Brazilian Amazonia. In this sense, our specific objectives were to briefly describe the sociodemographic profile of the participants and to investigate variables that contribute to better levels of subjective well-being (SWB) and stress of the participants, who were in remote education by digital means.

## 2. Materials and Methods

### 2.1. Characterization of the Research

This research has a quantitative approach [28], with an exploratory feature [29], carried out with children from 8 to 12 years old, students from private schools in the city of Santarém, State of Pará, who were in education through digital means as a result of the pandemic caused by COVID-19. Students and parents or legal representatives participated in the research voluntarily. Our research was carried out in private schools due to the stoppage of activities in public schools in the city, due to the pandemic, causing a lack of availability of teaching by digital means to students in the public network.

### 2.2. Sampling Criteria

The mapping of schools in the private network was carried out by searching the QEdu website (https://novo.qedu.org.br/, accessed on 10 February 2021), which extracts all data from the school census of the Ministry of Education of Brazil. The search found 81 private schools in Santarém. Of these, only 13 schools were operating during the pandemic, and so prior contact was established in order to obtain consent to carry out the research. There was only one school we could not establish contact with, even after several attempts. Furthermore, only one school refused to participate in the study, due to work overload; another school, after previous contact, did not confirm participation.

Then, letters were sent for participation and authorization of data collection from 10 schools. Subsequently, each school administration communicated to the parents about the existence and accomplishment of the research. Thus, in the end, six schools accepted and authorized their participation in this study, with a total of 1147 enrolled children. Two schools had a low number of children enrolled. Thus, these were combined into one total of 39 children.

Taking into account the population of 63,862 of children aged 8 to 12 years in Santarém [30], the formula *n* = N/4(N + 1)D + 1 was used in which: N = target-population’s size; D = ratio between B^2^/z^2^; B = maximum error (0.05); and z = deviation from the standard normal of 1.96 for 95% reliability [31].

Initially with the mentioned parameters, we arrived at the approximate sample value of 382 children, calculated by the free program RStudio^®^, version 1.2.1335 2019, an interface for the R program [32].

After verifying the number of students per school, the proportional stratified sample was calculated based on the population of 1147 students from the schools that gave consent for the research. With this, the approximate value of 317 children composing the research sample was reached.

However, as the study used an online survey with several instruments and several items to be answered for data collection, in the middle of a pandemic that led to intense use of technologies on a daily basis, there was a low adherence by students and their parents. Therefore, this study consisted of only 44 subjects who agreed to participate in the research.

### 2.3. Research Instruments

#### 2.3.1. Multidimensional Scale of Life Satisfaction for Children (MSLSC)

The Multidimensional Life Satisfaction Scale for Children (MSLSC) [33] was applied to the students because it is a reduced version of the original instrument. It is a 5-point Likert-type scale, which, due to restrictions arising from the pandemic, was applied via Google forms.

The scale has 32 items (Table A1), distributed in five dimensions, found in Table 1. To answer each question, the child marked from 1 to 5 what he felt about the question, where 1 means not at all, 2 little, 3 more or less, 4 quite a lot, and 5 very much. In the systematization of the answers, we inverted the values of the scores referring to the dimension Compared Self, as they have a negative meaning [33]: 1↔5; 2↔4; 3↔3; 4↔2; 5↔1, for items 12, 15, 19 and 22. Thus, if the child scored 5, 1 was added to the raw score and so on.

#### 2.3.2. Children’s Global Life Satisfaction Scale

The second instrument used was the Children’s Global Life Satisfaction Scale [34], with good psychometric consistency (α = 0.83). Cronbach’s alpha coefficient (α) is a way of estimating the reliability of an instrument. With it, it is possible to analyze the overall life satisfaction of the child. The scale has 7 items related to subjective well-being. Its application is simple and fast, consisting in assessing the life satisfaction of children from 7 years of age. It is presented as a 5-point Likert-type scale, where each item varies from 1 to 5 points. In the end, there is a sum that varies from 7 (low satisfaction) to 35 (high satisfaction) [35]. All the mentioned instruments were adapted for the Google Forms platform.

The MSLSC and the Children’s Global Life Satisfaction Scale are important instruments in measuring children’s SWB. They allow the observation of the child’s levels of satisfaction in a global or multidimensional way, providing data for suggestions for future interventions based on the analysis of each domain [33].

#### 2.3.3. Infantile Stress Scale

The Infantile Stress Scale is recommended to be applied with children from 6 to 14 years old. It aims to measure the child’s feelings in the face of disturbing situations in their daily lives [36]. The instrument has 35 items divided into circles with four equal parts. In the online format, this scale was arranged in four points to be scored according to the question.

In this way, as this child goes through some specific event, in each topic he/she marks a quadrant of that circle. That is, he/she can: not mark any quadrant, representing something that “never happens”; mark a quadrant, meaning that the event happens “a little”; two quadrants when she believes it happens “sometimes”; three quadrants when “almost always” occurs; and all circles can be described as “always happens”.

These 35 items, on a scale of 0 to 4 points, are grouped into four factors: Physical Reactions (PR), Psychological Reactions (PsR), Psychological Reactions with Depressive Component (PRDC), and Psychophysiological Reactions (PPR) [23]. They are randomly distributed, in which the PR factor includes items 2, 6, 12, 15, 17, 19, 21, 24 and 34; the PsR are in items 4, 5, 7, 8, 10, 11, 26, 30 and 31; the PRDC are found in items 13, 14, 20, 22, 25, 28, 29, 32, and 35; and finally, the PPR in items 1, 3, 9, 16, 18, 23, 27, and 33 (Table A2). Thus, it is possible to quantitatively analyze the stress phases that the child is in, be it in the alert, resistance phases, of near exhaustion or exhaustion.

In the first phase described, it is not yet considered a serious factor (Table 2). In the second phase, resistance, the child needs to expend more energy to go through the stress process when it occurs. In the third phase, the near-exhaustion phase, it is characterized by very severe stress in the child, who may become physically or psychologically ill; the exhaustion phase is considered the most serious, when the child gets sick [37].

### 2.4. Data Collection

The questionnaires’ link was sent to the directors/coordinators of the schools that authorized the participation. The single link contained a form with: the Free and Informed Consent Form (ICF) for the father, mother or legal guardian to fill out; the Free and Informed Assent Term (FIAT) for the child to complete; the sociodemographic questionnaire and the Children’s Stress Scale; the Children’s Multidimensional Life Satisfaction Scale; and the Children’s Global Life Satisfaction Scale.

The collection started on 31 May 2021. The school board shared the link, a presentation video and a short textual summary of the video directly to parents through the WhatsApp^®^ application. This stage was completed on 31 August 2021, after reaching the mark of 44 participants.

### 2.5. Data Analysis

Data were analyzed using descriptive statistics and simple logistic regression, which assumes that the response variable is distributed according to a binomial probability model. The logistic regression model is one of the generalized linear models (GLM) class, which is an extension of the normal linear models [38]. From this technique, the probability of occurrence of an event of interest can be predicted; in this case, the response variable (Y), from a covariate (X), also called the independent variable. In this case, the response variable is dichotomized, that is, it assumes only two values, usually 1 (one) or 0 (zero), to represent, respectively, the occurrence or not of the event of interest. Logistic regression was calculated using the equation: P(*Y*) = 1/(1 + e^(−(b_0_ + b_1_ X_1_))) where: P(*Y*) is the probability of event Y occurring, b_0_ and b_1_ are the parameters of the model to be estimated, where b_1_ is the value to be performed in the model hypothesis test to find out if X_1_ is relevant to the model [39].

In order to achieve our goal, when selecting the variables and organizing the best model, it was necessary to go through some adjustments. One of them was to verify how the predictor could be made accurate. Sensitivity would be the positive true values, evaluating how much the model could be a real predictor [40]. The formula is presented as SENS = TP/FN where: SENS means specific sensitivity; TP means True Positive values; and FN means False Negatives. Specificity, on the other hand, represents the value of true negatives, that is, verifying the amount of data that the predictor event may not occur. The calculation is given by SENS = TN/TN + FP where: VN = True Negatives; and FP = False Positives. Then, the confusion matrix was generated, explaining the predictor values with the result of the model’s precision, with any value of 74% and over considered as excellent accuracy [40].

For the Infantile Stress Scale, a variable called resistance 2 was created and it was dichotomized as follows: occurrence of resistance event 2 (one) or absence of this event (zero). That is, if the child’s stress phase is alert or resistance, then the resistance event 2 is considered to have occurred, and if the stress phase is normal, the event is considered not to have occurred.

Thus, it can be considered that the resistance 2 variable is a binomial distribution with “n” trials (number of interviewees) and success probability “*p*” with success being the occurrence of the resistance 2 event. The resistance 2 variable is then considered as our response variable (Y) and, thus, the probability of Y happening from the covariates of the instruments used in this research can be estimated. The GLM contains several adjustment options for situations in which the distribution of Y is not Normal, among which the logistic regression stands out, which fits perfectly into this design to estimate the probability of a child being in resistance phase 2 or not, from sociodemographic variables, class situation during the pandemic by digital means or other variables.

The processing of data and adjustment of the logistic regression model was carried out in the R program, using several packages, among which the glm function of the stats package [32] can be highlighted. Initially, the complete model was executed with the following variables: gender, age, children’s class format by digital means, domain levels of the MSLSC, and the Children’s Global Life Satisfaction Scale (CGLSS). With the stepwise algorithm of the step function, also from the *stats* package, it was verified that only the family variable presented statistical significance for the model. In this sense, a detailed analysis of this model was carried out.

Finally, we obtained the odds ratio of the data. The Odds Ratio (OR) places the model as an exponent through the function exp() [39], through R. When the value is greater than 1, it means that when the predictor increases, the chances of the event increase, but if the value is less than 1, as the predictor increases, the probability of the event happening decreases [39].

### 2.6. Ethical Aspects

This research was approved by the Research Ethics Committee of the State University of Pará (Campus XII—Tapajós) under protocol CAAE: 34151320.2.0000.5168. To ensure the anonymity of the participants, each school received a number, as well as the letter P of the participant and a numbering according to the order of each answer in the form.

Each subject (the child, as well as those responsible for them) was informed about the research, expressing their acceptance to participate in this investigation by signing a Free and Informed Consent Term (FICT).

## 3. Results

Forty-four students from private educational institutions participated in the research, who agreed to respond to the research instruments after authorization from the schools and parents/guardians. Among the children, 23 (52.3%) were female and 21 (47.7%) were male, with a mean of 10.5 years ± 1.38 years. It was found that 13 (29.5%) subjects were in their 7th year of Fundamental Level.

The CGLSS revealed scores between 27 and 34 points. In this situation, it is understood that the participants have a high level of life satisfaction, with a median of 4.6 points and an average of 4.33 points.

Data regarding the completion of the Multidimensional Scale of Child Life Satisfaction by domain were evidenced (Table 3). It was observed that the Family domain had the highest level of satisfaction with an average of 4.54 ± 0.45, followed by Friendship with an average of 4.09 ± 0.73.

For the Infantile Stress Scale, Psychophysiological reactions generate an average of 10.9, followed by Psychological reactions and Depressive Components, with averages of 9.50 and 8.20, respectively (Table 4).

Our results show that most children are in the normal phase (63.6%). However, 11.4% of the participants were in the resistance phase during the period of application of this instrument (Table 5). No children were observed in the phase of near exhaustion or exhaustion.

The Family dimension had a strong positive correlation (r = 0.70; *p*-value < 0.05) with the Self dimension (r = 0.70; *p*-value < 0.001) and moderate correlation with Psychophysiological reactions (r = 0.68; *p*-value < 0.001) (Table 6). Physical reactions showed a strong correlation with Psychophysiological reactions (r = 0.83; *p*-value < 0.01).

The complete model was adjusted, that is, with all covariables of interest. Subsequently, from the stepwise regression method of the step package, which combines the deletion of variables, we arrived at the most adequate model for the data according to the AIC criterion. In this model, the only predictor variable used was Family (*p*-value < 0.05). The fitted model (Table 7) (ŷ = 10.78 − 2.51 × FAMILY.DOMAIN) was significant (*p*-value = 0.00) and the negative β_1_ value indicates that as there is an increase in the mean of family domain responses, the probability of a child entering a resistance phase decreases (Figure 1).

The deviance analysis compared the null model (only with the intercept) with the new model (with the covariate Family domain), being significant (*p*-value = 0.00) with standardized residuals between −1.43 and 2.00 (Table 8), characterizing error independence and indicating that there is strong evidence that the Family domain model is better than the null model.

The proportion of correct answers in the model confusion matrix, using the *ClassLog* function [41], was 50% for sensitivity (proportion of true positives) with a specificity of 92.86% (proportion of true negatives) (Table 9). The proportion of correct predictions over the total, that is, the accuracy of the model, was 77.27%.

The exponential of the estimates (odds ratio) of the parameters of the fitted model showed a negative coefficient for the adjusted model, indicating that an increase in the Family domain score decreases the chances of the child being in resistance 2. As the answer is the logit function, by exp(β), there is a value lower than 1, confirming that the higher the value of the variable (Family domain), the lower the chances of a child belonging to the resistance phase 2 category. Thus, by each 1-point increase on the predictor, the chances of a child entering resistance phase 2 are reduced by around 92% (1 − 0.08) (Table 10).

## 4. Discussion

In our study, we analyzed the subjective well-being and stress of children aged 8 to 12 years who were studying by digital means due to the COVID-19 pandemic, in the city of Santarém, Brazilian Amazon. We used different scales to carry out this assessment.

A study carried out with 96 school-age children, also in the city of Santarém, Pará, used the Multidimensional Child Life Satisfaction Scale containing 50 items, analyzing five existing domains, plus the non-violence domain, having found that the average in the School domain was higher (4.07 ± 0.56) than the Family domain (3.94 ± 0.47) [42].

The current research, however, showed higher averages found in the Family (4.54 ± 0.45) and Friendship (4.09 ± 0.73) domains; it is possible to identify that the change of context of children may have influenced the increase in the average score in the Family domain, showing that living with family for a longer period can be beneficial for the participant. However, caution should be exercised, because in the research involving school-age children with digital teaching, the instrument used was the reduced version with 32 items. However, our results point to an important domain to be invested in during this difficult context of the pandemic.

Studies on children’s SWB cover the various domains of a child’s life. One of the delicate factors to address is the family. The family context in which the child is immersed can directly influence well-being. It is worth remembering that there is no research in the literature that proves the existence of an ideal family model. Thus, when comparing the SWB of Brazilian children aged 9 to 13 who have different family configurations, through the application of the Personal Well-Being Index-School Children, the Brief Multidimensional Student’s Life Satisfaction Scale and the General Domain Satisfaction Index, significant differences were observed in the SWB of children who have a complete family situation, compared to those who live only with a single father or mother, or in a couple that has other children and is multigender [41].

It is not our intention to discuss the ideal model for raising a child, but how beneficial a comforting family environment can be, since it provides high levels of infantile life satisfaction, having evidenced here a positive influence of the family on the development of these students.

One of the reasons that could explain the correlation of the Family domain with the Stress Scale would be the direct contact in the environment in which these children are immersed. In this case, the family is the first context to be experienced by the child, from birth and arriving at the school environment. In this way, family life can add positive values to child development [43].

The existence of a stable family environment in the children’s lives is a crucial factor for the balance in the relationship between them. Parents need to keep the emotional factor in equilibrium to enable a more harmonious coexistence in their own home, providing well-being [44].

However, if the family context is unstable, there may be a negative influence on the lives of these children. Behavioral changes in parents or guardians can be noticed by the children. Therefore, it is important that the family environment is stable to contribute positively to child development [45]. This may also show influence in the child’s own personality.

With the intention of studying psychological symptoms and the COVID-19 pandemic, an investigation was carried out to verify symptoms related to suicidal ideation, self-injury and suicide attempts in 1241 children and adolescents between 9 and 15 years of age before the first and after the second wave of the pandemic [46]. The authors showed an increase in depressive symptoms [OR 1.50; 95% CI, 1.18–1.90], in non-suicidal self-harm [OR 1.35; 95% CI 1.17–1.55], in suicidal ideation [OR 1.32; 95% CI, 1.08–1.62] and in suicide attempts [OR 1.71; 95% CI, 1.31–2.24] during the second wave.

In India, researchers studied children and adolescents aged 9 to 18 years quarantined in the family context due to the COVID-19 pandemic, aiming to describe the psychological impacts during this period [26]. The researchers divided 131 participating children and adolescents into two groups: a quarantined group; and a non-quarantined group. It was evidenced that 68% of the participants felt helpless and 61.98% felt afraid during the quarantine, and the conditions of fear (*p* ˂ 0.0001), nervousness and annoyance affected the group that was not quarantined [46]. This shows that the actions of the external environment can cause physical and physiological impacts through reflexes generated in the psychological field.

In research carried out in South Korea to verify the SWB and stress of 166 participants evaluated before the COVID-19 pandemic and during the pandemic, stress and conversation time with parents were analyzed as variables [47]. The children in the group during the pandemic had high rates of conversation time with their parents during this period; however, the stress levels of these subjects were higher during the pandemic period [47]. This research corroborates our findings, as the family is an important variable in times of high stress in this community. Therefore, the family situation becomes a possible support factor in relief from these symptoms.

Thus, we can conclude the presence of the family in a child’s life becomes essential, as was seen with the participants of the research carried out in Santarém, where with each point increase in the Family domain score, the chances of these subjects entering a resistance phase decreased.

In this way, another study was carried out, also online, based on reports from family members on Facebook, analyzing how parents/guardians are dealing with the education of children at home during the pandemic, revealing that the family has a pleasant participation in the life of a school-aged child [48]. Most of the reports showed how parents were able to accompany their children or grandchildren in online classes, putting them in a study routine without losing the content and performing well even in education by digital means [48].

Thus, the family can positively influence the well-being and development of the child. These subjects, in the presence of their parents or their closest guardians, may feel more comfortable in their learning, as it is a family member who teaches them or helps them with homework. During the pandemic, this coexistence became the “new normal” for many people.

The family context can have a negative impact on the lives of children who spend more time at home than at school. Families with greater socioeconomic vulnerability are more likely to experience an increase in depressive and anxiety symptoms, and increased cases of domestic violence may be observed [7]. The authors also state that the pandemic situation should bring hyper-vigilance on the part of parents because of the excess of bad news associated with the growing number of deaths.

In this line, the study of child stress is fundamental and can evidence the existence of children in a phase of alert or resistance. In this case, staying at home for a long time, due to isolation and social distancing, may be a generating factor for these findings. This means that children who do not have a family circle with many people around them can feel lonely and have their quality of life and well-being affected. In the case of the alert phase, it may be that the participants have some kind of fear, either by being alone or fear of contracting the disease. However, this fear can have a positive perspective as a protective factor for the child and for family members, and attention should be paid to the feeling of loneliness, which can lead to symptoms such as anxiety and depression [49]. Problems related to the worsening mental health of parents (26.9%) and the behavior of children (14.3%) were found from March to June 2020 in the United States [50].

In this case, the positive influence of the family on children’s mental health status may reflect the good psychological health of the parents/guardians themselves. If the opposite was the case, there would probably be a drop in the scores of the Family domain in the MSLSC and an increase in the levels of stress in the participants, showing the importance of parents’ mental health to maintain the well-being of their children.

The COVID-19 pandemic can affect the well-being of children and families, but to overcome this socially and culturally complex period, the state must invest in family support services, including those dealing with mental health and education, in search of strong and healthy family relationships [51].

## 5. Conclusions

This research brought important elements to the study of children isolated because of the COVID-19 pandemic in education by digital means. When verifying the children’s subjective well-being correlated with the participants’ stress levels, the importance of the family context was revealed during a longer time spent at home, and without contact with the school. In view of this, it was evidenced that SWB and childhood stress are positively influenced by the family in the child’s life, due to the long time spent at home. In this way, it is possible to verify that when the scores of the research instruments increase, the chances of children getting high levels of stress due to family support in the context of the COVID-19 pandemic decrease. This study may serve as a basis for further investigations into stress-related childhood SWB. Therefore, research of this nature that involves social isolation, restriction of access to open places and the realization of classes with school activities by digital means should be encouraged and carried out, so that there can be a better confrontation of the obstacles related to the pandemic and epidemic diseases, ensuring that they affect as little as possible the well-being of the population.

The sanitary restrictions of the pandemic made it impossible to carry out this study in person, which may have affected the responses given by the participants. The undertaking of this research only in private schools constitutes another limitation since the public schools in the city of Santarém had suspended teaching activities. However, this gap may provide insights for future research. Conducting this survey online may have been the cause of low adherence by the students’ parents. It remains necessary to carry out more studies with larger samples, involving more schools, including public schools.

The family revealed its role in managing stress and maintaining the well-being of children in education by digital means. This implies that the family and its members recognize their roles and identify ways to improve their relationship with their children, aiming at their well-being. It is also up to schools to identify ways to cooperate with families to prioritise children’s quality of life.

## Figures and Tables

**Figure 1 ijerph-19-07148-f001:**
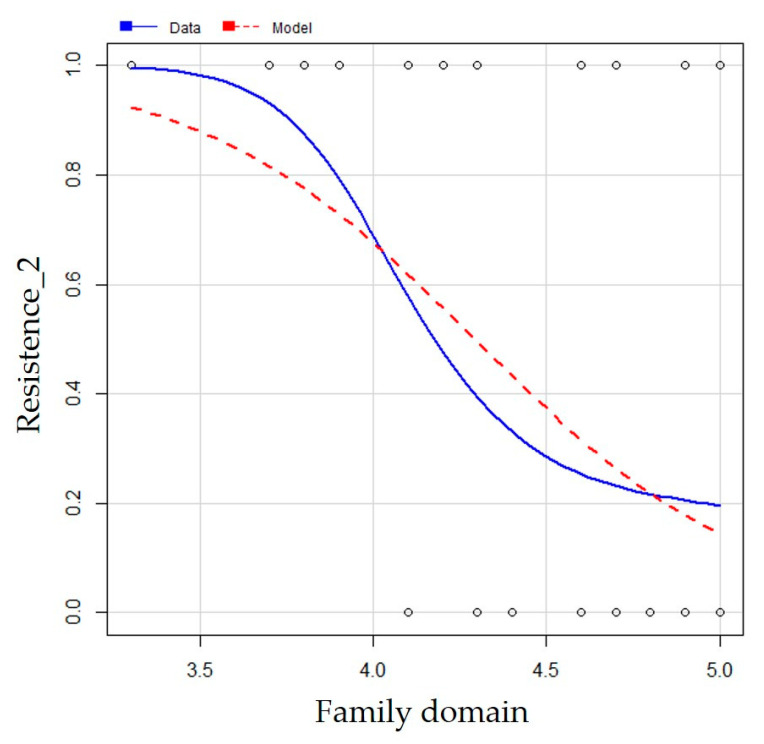
Observed data of the variable resistance_2 and fitted data of the model. Source: Data Bank of this Research.

**Table 1 ijerph-19-07148-t001:** Items of the dimensions of the MSSLC.

Dimension	Number of Items	Items
Family	9	2, 6, 9, 13, 16, 20, 23, 26, 32
Self	8	1, 5, 8, 18, 21, 24, 25, 29
Friendship	7	3, 7, 10, 14, 17, 27, 30
School	4	4, 11, 28, 31
Compared Self	4	12 *, 15 *, 19 *, 22 *

Source: Oliveira et al. [33]. This Journal is based on the Budapest Open Access Initiative, copyright is guaranteed when the source is cited. * Items that should have their scores inverted before the calculation of the raw score.

**Table 2 ijerph-19-07148-t002:** Criteria to determine the phases of the Infantile Stress Scale.

Phases of Stress	Criteria
Not considered grave	Sum lower than 39 points on the total score of the scale
Alert phase	Sum higher than 39 points on the total score of the scale
Resistance phase	Sum from 59 to 79 points on the total score of the scale
Near exhaustion phase	Sum from 79 to 99 points on the total score of the scale
Exhaustion	Sum higher than 99 points

Source: Lima [37]. The copyright is guaranteed when the source is cited.

**Table 3 ijerph-19-07148-t003:** Dimensions of the MSSLC, in the context of the COVID-19 pandemic, of students from the 2nd year to the 9th year of the Basic Level of private schools in Santarém, Pará, Brazil.

Measure	Dimensions				
	School	Family	Friendship	Self	Compared Self
Mean	3.99	4.54	4.09	3.97	3.27
Median	4.25	4.67	4.29	4.13	3.25
Standard Deviation	0.95	0.45	0.73	0.69	0.92
1st Quartile (25%)	3.44	4.33	3.71	3.63	2.69
2nd Quartile (50%)	4.25	4.67	4.29	4.13	3.25
3rd Quartile (75%)	4.75	4.89	4.61	4.53	3.75

Source: Data Bank of this Research.

**Table 4 ijerph-19-07148-t004:** Infantile Stress Scale per reaction in the context of the COVID-19 pandemic, of students from the 2nd year to the 9th year of the Basic Level of private schools in Santarém, Pará, Brazil.

Statistics	Reactions			
	Physical	Psychophysiological	Depressive Component	Psychological
Mean	5.93	10.9	8.20	9.50
Median	5.50	11.0	7.50	8.00

Source: Data Bank of this Research.

**Table 5 ijerph-19-07148-t005:** Total frequency of the Phases of Stress in the participants, students from the 2nd year to the 9th year of the Basic Level of private schools in remote education in Santarém, Pará, Brazil.

Phases of Stress	Relative Frequency	Absolute Frequency
Normal	28	63.6%
Alert	11	25.0%
Resistance	5	11.4%
Total	44	100%

Source: Data Bank of This Research.

**Table 6 ijerph-19-07148-t006:** Correlations between the Infantile Stress and Multidimensional Life Satisfaction Scales.

Scales	Statistics	Family Dimension	Self Dimension	Physical Reactions	Psychophysiological Reactions
Family Dimension	r-Pearson	-----			
	value *p*	-----			
Self Dimension	r-Pearson	0.70	-----		
	value *p*	< 0.001	-----		
Physical Reactions	r-Pearson	−0.55	−0.25	-----	
	value *p*	<0.001	0.104	-----	
Psychophysiological Reactions	r-Pearson	−0.68	−0.42	0.83	-----
	value *p*	<0.001	0.004	<0.01	-----

Source: Data Bank of this Research. Shapiro–Wilk normality test: Family Dimension: *p* < 0.001. Self Dimension: *p* = 0.006. Physical Reactions: *p* = 0.069. Psychophysiological Reactions: *p* = 0.182.

**Table 7 ijerph-19-07148-t007:** Estimates of the fitted model parameters.

	Estimate	Standard Error	z Value	*p*-Value
Intercept (β_0_)	10.78	4.02	2.69	0.00
Family Domain (β_1_)	−2.51	0.89	−2.83	0.00

Source: Data Bank of this Research.

**Table 8 ijerph-19-07148-t008:** Deviance analysis comparing the null model (only with the intercept) with the new model (with the co-variable Family domain).

	Liberty Degrees	Residual Deviance	*p*-Value
Null model	43	57.68	-
New Model	42	47.33	0.00

Source: Data Bank of this Research.

**Table 9 ijerph-19-07148-t009:** Proportion of successes of the model confusion matrix.

Estimated Value	Observed Value		Precision
	Y = 0	Y = 1	
Y = 0	26	8	-
Y = 1	2	8	-
Correct classifications	0.9286	0.5000	0.7727

Source: Data Bank of this Research.

**Table 10 ijerph-19-07148-t010:** Exponential of the estimate (odds ratio (OR)) of the parameters of the fitted model.

	Odds Ratio (OR)	Interval of Trust (95%)	
		Inferior Limit	Superior Limit
exp (β_0_)	4.85 × 10^4^	18.42	1.27 × 10^8^
exp (β_1_)	0.08	0.01	0.05

Source: Data Bank of this Research.

## Data Availability

The data presented in this study are available in the text, figures, and tables.

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
