# Peer review of "Well-Being and Stress of Children in Teaching by Digital Means during the COVID-19 Pandemic: A Case in Santarém, Brazilian Amazonia"

_ijerph, 2022, doi:10.3390/ijerph19127148_

Round 1
Reviewer 1 Report
Although the sample is small, the study is interesting and the results are well presented. However, further elaboration of the conclusions is recommended. It is recommended to expand and deepen them.
Author Response
Although the sample is small, the study is interesting and the results are well presented. However, further elaboration of the conclusions is recommended. It is recommended to expand and deepen them.
Response: We adjusted the conclusion (Lines 516-543).
Reviewer 2 Report
Very interesting topic under study approached from a rigorous and systematic research procedure.
The following aspects need to be improved:
- It is necessary to specify a main objective with its specific sub-objectives that derive from it.
- In the introduction it is necessary to include the state of the question on the object of study.
- In the discussion section, it is necessary to reformulate to focus in detail on the contrast of the results presented with the previous investigations (explained in the state of the question).
- In the conclusions section, it is necessary to reformulate to focus on the most relevant conclusions in response to the objectives set based on the results obtained and analyzed. It would also be interesting to delve into the practical implications of the research carried out as well as the limitations and future research proposals.
Author Response
- It is necessary to specify a main objective with its specific sub-objectives that derive from it.
Response: We have adjusted the objectives (Lines 149-154)
- In the introduction it is necessary to include the state of the question on the object of study.
Response: We made changes to the Introduction (Lines 83-105; 116-144).
- In the discussion section, it is necessary to reformulate to focus in detail on the contrast of the results presented with the previous investigations (explained in the state of the question).
Response: we present details of some studies (Lines 465-472).
- In the conclusions section, it is necessary to reformulate to focus on the most relevant conclusions in response to the objectives set based on the results obtained and analyzed. It would also be interesting to delve into the practical implications of the research carried out as well as the limitations and future research proposals.
Response: We've made adjustments this section (Lines 516-548). Thank you
Reviewer 3 Report
During these last two years, an unprecedented series of restrictive measures have been established throughout the planet with the aim of containing the COVID-19 pandemic. Throughout these 24 months, various studies have been carried out in which it is pointed out that these restrictions could be evolving to the mental health of many people. Specifically, they could be causing psychological symptoms such as anxiety, stress, and depression (Wang, Pan, Wan, Tan, Xu, Ho, & Ho, 2020).
The results of some show how quarantine can be an unpleasant experience due to separation from loved ones, loss of freedom, uncertainty about the state of the disease and/or boredom (Brooks, Webster, Smith, Woodland, Wessely , Greenberg and Rubin, 2020); but most of these studies do not focus on the child population. In this sense, the present investigation is interesting, since young children have experienced this restrictive situation in their homes; therefore, the potential psychological affectation derived from confinement during the COVID-19 pandemic becomes an important objective to study. Thus, this research work aims to analyze the subjective well-being and stress of children aged 8 to 12 years in digital education during the Covid-19 pandemic.
Although the theoretical and methodological approach of this research work is correct, there are a number of issues that could help improve it and which are detailed below:
a) Aspects related to formal issues:
1. Uniformity with the acronyms of the variables studied would be necessary. In the text, the same variable is labeled as BES and as BS: Subjective Well-being
2. The data on the number of participants in the sample can mislead the reader. It seems that the figure "317" is given more importance than "44", the latter being the one that represents the final sample of participants.
3. The age of the participants should be indicated with the mean and standard deviation (SD)
b) Content-related aspects:
1. The theoretical foundation on the two variables studied - Subjective well-being and Stress - is scarce.
2. The fact that the sample was collected only from private schools is a limitation and bias important enough to address in more than just the final sentence of the study.
3. In the "Discussion" of this research work it is stated:
The existence of a healthy family environment in the lives of children is a crucial factor for the balance in the relationship between them.
It should be clarified, from a "scientific" point of view, what is meant by "healthy", because we could "fall" into moral questions that are irrelevant when we are doing research.
In this regard, the following is also stated:
However, if the family context is unstable, there may be a negative influence on the lives of these children.
Was "unstable" intended to be used as opposed to "healthy"? This aspect should be further nuanced.
4. The instruments used in this study are three. Two of them, apparently, to evaluate the same variable (Subjective Well-being): Children's Global Life Satisfaction Scale and the Multidimensional Life Satisfaction Scale for Children (MSLSC).
I believe that the reason for using two scales to apparently measure the same thing should be justified.
c) Aspects related to improvement proposals:
1. The differences between the participants in the sample could have been analyzed in terms of different sociodemographic variables such as gender, age, etc... in order to establish profiles of those who have suffered higher levels of stress, for example.
Author Response
a.1. Uniformity with the acronyms of the variables studied would be necessary. In the text, the same variable is labeled as BES and as BS: Subjective Well-being
Response: we have made adjustment in the text.
a.2. The data on the number of participants in the sample can mislead the reader. It seems that the figure "317" is given more importance than "44", the latter being the one that represents the final sample of participants.
Response: We did correct (Line 193-196). Thank you
a.3. The age of the participants should be indicated with the mean and standard deviation (SD)
Response: we did adjust (Line 332)
b.1 The theoretical foundation on the two variables studied - Subjective well-being and Stress - is scarce.
Response: We did improve (Line 83-105; 116-144)
b.2 The fact that the sample was collected only from private schools is a limitation and bias important enough to address in more than just the final sentence of the study.
Response: We did insert this information also in Lines 164-166
b.3. In the "Discussion" of this research work it is stated:
The existence of a healthy family environment in the lives of children is a crucial factor for the balance in the relationship between them.
It should be clarified, from a "scientific" point of view, what is meant by "healthy", because we could "fall" into moral questions that are irrelevant when we are doing research.
In this regard, the following is also stated:
However, if the family context is unstable, there may be a negative influence on the lives of these children.
Was "unstable" intended to be used as opposed to "healthy"? This aspect should be further nuanced.
Response: Corrected to "stable", referring to homes with less conflict and tension (Line 440), removed "healthy" (Line 446)Corrected to "stable", referring to homes with less conflict and tension (Line 440), removed "healthy" (Line 446)
b.4. The instruments used in this study are three. Two of them, apparently to evaluate the same variable (Subjective Well-being): Children's Global Life Satisfaction Scale and the Multidimensional Life Satisfaction Scale for Children (MSLSC).
I believe that the reason for using two scales to apparently measure the same thing should be justified.
Response: we justify in the methodology (Lines 228-231)
c.1. The differences between the participants in the sample could have been analyzed in terms of different sociodemographic variables such as gender, age, etc... in order to establish profiles of those who have suffered higher levels of stress, for example.
Response: We insert the explanation (Lines 366-370)